# EMHD Nanofluid Flow with Radiation and Variable Heat Flux Effects along a Slandering Stretching Sheet

**DOI:** 10.3390/nano12213872

**Published:** 2022-11-02

**Authors:** Aamir Ali, Hajra Safdar Khan, Salman Saleem, Muhammad Hussan

**Affiliations:** 1Department of Mathematics, COMSATS University Islamabad, Attock Campus, Kamra Road, Attock 43600, Pakistan; 2Department of Mathematics, College of Science, King Khalid University, Abha 61413, Saudi Arabia; 3Department of Mathematics, Government College University, Faisalabad 38000, Pakistan

**Keywords:** EMHD, nanofluid, variable thickness, non-uniform heat flux, thermal radiation, viscous dissipation

## Abstract

Nanofluids have gained prominence due to their superior thermo-physical properties. The current paper deals with MHD nanofluid flow over a non-linear stretchable surface of varying thickness in the presence of an electric field. We investigated the effects of nanometer-sized copper (Cu) particles in water (base fluid) as a nanofluid, as well as non-linear thermal radiation, variable fluid viscosity, Joule heating, viscous dissipation, and non-uniform heat flux. The current study’s aim is influenced by the immense applications in industry and machine building. It has been observed that linear stretching sheets have been extensively used in heat transfer research. Moreover, no effort has been made yet to model a non-linear stretching sheet with variable thickness. Furthermore, the effects of electromagnetohydrodynamics (EMHD) boundary-layer flow of a nanofluid with the cumulative impact of thermal radiation, variable viscosity, viscous dissipation, Joule heating, and variable heat flux have been investigated. Sheets with variable thicknesses are practically significant in real-life applications and are being used in metallurgical engineering, appliance structures and patterns, atomic reactor mechanization and paper production. To investigate the physical features of the problem, we first examined the model and identified all the physical properties of the problem. This problem has been formulated using basic laws and governing equations. The partial differential equations (PDEs) that govern the flow are converted into a system of non-dimensional ordinary differential equations (ODE’s), using appropriate transformations. The Adam–Bashforth predictor-corrector technique and Mathematica software are utilized to numerically solve the resulting non-dimensionalized system. The interaction of various developing parameters with the flow is described graphically for temperature and velocity profiles. It is concluded that the velocity of nanoparticles declines as the intensity of the magnetic field increases. However, the temperature of the nanomaterials rises, as increasing the values of the electric field also increases the velocity distribution. The radiation parameter enhances the temperature field. The temperature of the fluid increases the occurrence of space- and time-dependent parameters for heat generation and absorption and radiation parameters.

## 1. Introduction

In recent years, different areas of technology have incorporated the use of fluid flow across a stretching sheet due to its various advantages. This led to an increased interest from scientists and researchers. The fluid flow on a continuous solid surface was first studied by Sakiadis [1] in comparison to a surface with a finite length. He found that the differential boundary-layer equations are applicable, provided that boundary-layer conditions are different in both cases. He analyzed fluid flow on a continuous surface and derived equations for the differential and integral momentum. Crane [2] expanded the work of Sakiadis by looking at fluid flow through a stretched plate with velocity related to its distance from the slit. By incorporating the influence of viscous dissipation and internal heat generation and absorption, Vajravelu and Rollins [3] studied the problem of heat transfer in laminar non-Newtonian fluid flow that passed through a stretched surface. Vajravelu and Cannon [4] investigated flow on a boundary layer for fluid passing through a non-linear stretched sheet. After the initial work of Sakiadis, several investigations have been conducted into the flow through a stretched surface for different flow configurations and fluid natures. Sheets with variable thicknesses are practically significant in real-life applications and are being used in metallurgical engineering, appliance structures and patterns, atomic reactor mechanization, and paper production. Lee [5] utilized a thin needle and assessed its flow over a boundary layer by comparing its thickness with that of the boundary layer. Ultimately, he deduced that in the case of a thin needle, the drag force per unit length and displacement thickness gradually decrease, but finally return to zero as the needle’s thickness becomes zero. Gupta and Gupta [6] used a stretched flat porous surface with uniform thickness and investigated the boundary flow over it. By analyzing two separate cases of CST (plates via continuous surface temperature) and PST (plates via prescribed surface temperature), Cortell [7] numerically investigated the transmission of heat in a viscous fluid via a non-linear stretched plate using the R-K scheme. Fang et al. [8] explored the flow of a boundary layer on an impermeable surface through a stretching plate with non-uniform thickness. By using a continuous impermeable stretched surface, Khader and Megahed [9] examined 2D Newtonian fluid flow and concluded that the local skin friction co-efficient diminishes with a rise in the slip velocity parameter. Elbashbeshy et al. [10] considered a visco-elastic fluid (Maxwell) and studied its behavior on a slandering stretching plate. Recently, Ali et al. [11] explored the effect of MHD Jeffery nanofluid flow via an elongating surface with various thicknesses.

Over the previous decades, several researchers have been captivated by the notion of nanofluids due to their improvement in the transfer of thermal energy. Choi [12] defined nanofluids as a colloidal solution of nano crystals and base fluids with diameters ranging from 1 nm to 100 nm and base fluids that are primarily water, ethanediol and lubricants. Nanofluids are used in electronic engineering, solar radiation, thermonuclear energy, heat exchanger tubes, medicinal products, coolant systems, and other fields due to their wide range of applications. Based on the consideration of the stretching plate, Khan and Pop [13] first anatomized nanofluid flow by applying Buongiorno’s model. Nadeem and Lee [14], influenced by thermophoresis and Brownian motion, modelled the flow of a boundary layer across a porous stretching sheet and employed HAM to find its solution and discussed heat and mass transfer analysis. Malvandi et al. [15] investigated the problem of unsteady nanofluid flow close to the stagnation point via a stretched surface under the influence of slip velocity and stretching parameters. An implicit finite difference technique was implemented by Khan et al. [16] to explore the problem of 3D nanofluid flow on an exponentially stretched sheet under the impact of thermophoresis and Brownian motion. Colangelo et al. [17] examined the relationship between microwaves and nanofluids. Maleki et al. [18] considered non-Newtonian (pseudo-plastic) nanofluid flow along a permeable surface and utilized the Runge–Kutta–Fehlberg fourth–fifth method to find the numerical solution. Lacobazzi et al. [19] investigated the connection between the clustering phenomena and the thermal conductivity of nanofluids. They also examined the impact of nanofluid stabilization on thermal conductivity. Ali et al. [20] discussed heat and mass transfer analysis across an exponentially extending surface for a 3D Maxwell nanofluid. After taking into consideration three kinds of nanoparticles, i.e., aluminum oxide, titanium dioxide, and iron oxide, Bognar et al. [21] discussed boundary-layer flow through a stretched plate. Ali et al. [22] recently published a paper on the 3D Oldroyd-B nanofluid across an exponentially extending surface. By comparing the performance of two distinct nanoparticles, i.e., zirconium dioxide-engine oil and copper-engine oil, Alazwari et al. [23] used the Keller-Box scheme to assess entropy production for non-Newtonian nanofluids over a porous surface and to find an approximate solution to the problem. They demonstrated that entropy elevates the Deborah number and that copper engine oil has a better value of thermal conductivity than zirconium dioxide engine oil. Aouinet et al. [24] examined the effects of the concentration of three nanoparticles, i.e., zinc oxide–water, silicon dioxide–water, and titanium dioxide–water, on velocity, temperature profiles, wall shear stress, and turbulence intensity. They demonstrated that while raising the nanoparticle concentration would increase the thickness, increasing the velocity causes the boundary layer thickness to drop. Waqas et al. [25] investigated Reiner–Philipoff nanofluid flow with a heat source/sink, melting phenomenon and thermal conductivity with the existence of motile microorganisms through a stretching surface. They showed that an increase in the Prandtl number reduces the thickness of the thermal boundary layer and increases the mass transfer of microorganisms via Peclet and bio-convection Lewis numbers. Imran et al. [26] studied the two-dimensional bio-convectional flow of cross nanofluids through a cylinder that contained motile microorganisms, while taking into account the effects of radiation, thermal conductivity, Brownian motion, and thermophoresis with the melting phenomenon. They found that a high Peclet number and bio-convection Lewis number lessen the microorganism profiles, but enhance them for high melting parameter values. By considering the impact of Cattaeneo–Christov and convective conditions, Rasool and Wakif [27] analyzed second-grade flow of nanofluids over a Riga plate and used the spectral local linearization method to find the solution to the problem. Colangelo et al. [28] performed experimental measurements and showed that nanofluids enhance the rate of heat transfer.

MHD flow in nanofluids has earned popularity due to its excellent ability to control the heat transfer rate. It is defined as the study of fluids that conduct electricity, encompassing the detailed study of their magnetic properties. Magneto-hydrodynamics (MHD) is made up of the following three terms: magneto (magnetic intensity), hydro (water) and dynamics (in motion) that were coined by Alfvin [29]. The behavior of magneto-hydrodynamic (MHD) flow near a stagnation point on a stretched plate was reported by Ishak et al. [30] and numerically evaluated by using the finite difference method. While considering a nanofluid flowing over a stretching plate, Ibrahim and Shankar [31] utilized Buongiorno’s model to study MHD flow with slip conditions. A time-independent, 2D magneto-hydrodynamic flow of Williamson fluid through a stretching surface with variable thickness has been scrutinized by Babu and Sandeep [32], by reviewing the effects of slip velocity, the Soret number, and the Dufour number. By using a time-independent nanofluid passing over a stretchable surface, Daniel et al. [33] studied MHD flow in the presence of an electric field under the influence of several parameters. Adem [34] addressed heat and mass transfer in a porous medium of MHD boundary-layer flow on a stretchable plate in the presence of an electric field. Ali and Ghori [35] examined the melting effects of Cattaneao–Christove and heat radiation on MHD nanofluid flow, including microorganisms. Souayeh et al. [36] discussed the flow of hybrid nanofluids through a peristaltic channel under the influence of EMHD, activation energy, radiation and gyrotactic microorganisms.

Thermal engineering’s subspeciality of heat transfer concerns the utilization, transformation, production, and transmission of heat energy between systems. There are several heat transfer mechanisms, including radiation, convection, and conduction. Heat transfer has numerous applications in climate engineering, chemical process industries, architecture, greenhouse effects and human body heat transfer. Temperature gradients have a large influence on fluid properties. Temperature and viscosity, in particular, have a direct relationship with gases, whereas temperature and viscosity have an inverse relationship with liquids. Radiation is a method or mode of transferring energy from one medium to another without the use of a third medium. Radiation from radioactive elements is emitted in all directions and travels to the point of absorption. Radiation can take the form of waves or particles. Several researchers are fascinated by the effect of thermal radiation on account of its extensive utilization in the engineering industry, i.e., as an electrical energy supply and in solar panels, atomic power stations, turbines. When a material is heated, radiation is emitted from the surface in the form of electro-magnetic waves, which is thermal radiation. All bodies emit thermal radiation with a temperature greater than zero. Abel and Mahesha [37] discussed the analysis of heat transfer while taking into account PST and PHF. Mukhopadhyay [38] studied the impact of thermal radiation and slip velocity on an exponentially stretched surface for MHD flow through a porous medium. While taking into account the two different cases, the PST case and the PHF case, Ahmad et al. [39] found an analytical solution via HAM of magneto-hydrodynamic fluid flow on an exponential stretched plate with radiation effects and noticed that the thermal boundary layer decreases with an increment in the Prandtl number. Under the influence of Brownian motion, thermophoresis, and radiation, Sheikholeslami et al. [40] considered two rotatable plates and analyzed the magneto-hydrodynamic flow of nanofluids between them. A reduction in the thermal boundary layer was observed because of the escalation in radiation parameters. By considering two types of nanofluids, aluminum copper–water and copper–water, Krishna et al. [41] analyzed the fluid flow through a slandering stretched sheet with slip effects. Khan et al. [42] showed the visco-elastic (Maxwell) nanofluid flow on a stretched plate caused by the effects of radiation and EMHD. Jamshed et al. [43] presented the analysis of entropy generation for an unsteady MHD flow of Casson nanofluid across a stretching sheet with slip conditions. They observed that elevating the Reynolds number, together with the Brinkmann number, improved the system’s overall entropy. Waqas et al. [44] explored the mathematical modeling of Carreau–Yasuda nanofluid flow on a porous surface in the presence of motile microorganisms and developed a numerical solution via the shooting method. They determined that an elevating bio-convection Lewis number and Peclet number decreases the microorganisms’ profiles, whereas boosting the microorganisms’ Biot number improves the microorganisms’ profile. Kumar et al. [45] explored tangent hyperbolic nanofluid flow over a stretching sheet with thermal radiation, MHD, a heat source, and convective boundary conditions. They used MATLAB bvp4c to develop a numerical solution.

The heat produced by the flow of an electric current through a conductor is referred to as Joule heating. Joule heating is used in a variety of everyday applications, such as electrical fuses, the glowing filament of an incandescent light bulb, electrical tabletop hotplates, and so on. Hsiao [46] analyzed EMHD-free convection and mass transfer nanofluid flow for a two-dimensional and incompressible fluid under the impact of a heat source/sink, viscous dissipation, radiation, thermophoresis, and Brownian motion. In order to develop an analytical solution, Sharma and Gupta [47] studied the flow of MHD nanofluids with heat flux and used HAM to account for the effects of the radiation parameter and viscous dissipation. Shahzad et al. [48] scrutinized the MHD Jeffery nanofluid boundary-layer flow in response to the consequences of viscous dissipation and ohmic heating on a stretching surface. They found that lowering the magnetic parameter lowered both the Nusselt number and the skin friction coefficient. Kumar et al. [49] expanded their work on Oldroyd-B nanofluids that pass over a stretched plate in light of Brownian and thermophoresis diffusion, viscous dissipation, thermal radiation, and ohmic heating effects. Muhammad et al. [50] used HAM to find a series solution and assessed the 3D magnetohydrodynamic flow generated by an exponentially stretched surface, after reviewing the impact of viscous dissipation and Joule heating. Maleki et al. [51] proposed the boundary-layer flow of nanofluids with heat generation and absorption through a permeable surface under the effects of thermal radiation, viscous dissipation, and slip conditions. Maleki et al. [52] carried out heat transfer analysis of pseudo-plastic nanofluids that flowed over a moveable porous surface with viscous dissipation and heat generation/absorption. They presented a numerical solution by utilizing the Runge–Kutta–Fehlberg fourth–fifth order. Zeeshan et al. [53] investigated the entropy generation and EMHD using Poiseuille flow of a nanofluid through a wavy channel, while accounting for thermal radiation, viscous dissipation, and Joule heating. They showed that increasing the pressure gradient causes an average energy loss. Thiagarajan and Kumar [54] proposed MHD boundary-layer flow of a nanofluid with heat production/absorption through an exponential stretched plate by considering Brownian and thermophoresis diffusion phenomena. Swain et al. [55] developed a heat transfer model for magnetohydrodynamic boundary-layer flow on a permeable surface that included Joule heating and viscous dissipation. Thumma and Mishra [56] found an analytical solution for the heat transfer analysis in an Eyring–Powell nanofluid under the effects of ohmic heating, heat flux, and viscous dissipation over an elongated sheet, by using the Adomian decomposition technique and came to the conclusion that enhancing the heat flux parameter actually reduces the shear stress rate. Sharma et al. [57] carried out heat and mass transfer analysis of magnetohydrodynamic fluid flow using HAM by incorporating the effects of viscous dissipation, radiation, chemical processes, variable viscosity, and heat sources through a stretched surface of varying thickness. Sajid et al. [58] addressed the heat and mass transfer analysis of an incompressible micro-polar fluid through a porous stretchable sheet and utilized the non-linear shooting method by considering viscous dissipation, thermal radiation, and convective boundary conditions. They noticed that higher order reactivity improves the heat transfer rate, whereas mass convection boosts concentration. In addition to analyzing the generation of flow in a parabolic trough solar collector through a non-linear stretching surface under the influence of radiation and Cattaneo–Christov heat flux, Abu-Hamdeh et al. [59] determined the entropy generation in a Powell–Eyring nanofluid across a porous surface. They observed that the generation of entropy is inversely related to thermal efficiency. Nazeer et al. [60] analyzed the mathematical modeling of non-Newtonian cross-flow with viscous dissipation and slip boundary conditions and presented the numerical solution of the problem.

The current study’s aim is influenced by the above-mentioned analysis and extensive applications in industry and machine building. It has been observed that linear stretching sheets have been extensively used in heat transfer research. Moreover, no effort has been made yet to model a non-linear stretching sheet with variable thickness of electromagnetohydrodynamics (EMHD) boundary-layer flow of a nanofluid, with the cumulative impact of thermal radiation, variable viscosity, viscous dissipation, Joule heating, and variable heat flux. Sheets with variable thickness are practically significant in real-life applications and are being used in metallurgical engineering, appliance structures and patterns, atomic reactor mechanization, and paper production. The purpose of this research is to examine the effects of electro-magnetohydrodynamics (EMHD), non-linear thermal radiation, ohmic heating, viscous dissipation, and variable heat flux on nanofluid flow over a slandering stretching plate. To investigate the physical features of the problem, we first examined the model and identified all the physical properties of the problem. This problem has been formulated using basic laws and governing equations. The prevailing equations of the flow model are highly non-linear PDEs (partial differential equations) that have been converted into a dimensionless form using similarity transformations and non-dimensional variables. The resulting non-dimensional system of equations has been numerically computed by utilizing the Adam–Bashforth predictor-corrector technique. Graphs were used to visually examine the effects of a variety of dimensionless parameters on the flow field. It is concluded that the velocity of nanoparticles declines as the intensity of the magnetic field increases. However, the temperature of nanomaterials rises, as increasing the values of the electric field also increases the velocity distribution. The radiation parameter enhances the temperature field. The temperature of the fluid increases the occurrence of space and time-dependent parameters for heat generation, absorption, and radiation. To the best of the author’s knowledge, this is the first attempt to investigate the EMHD boundary-layer flow of a nanofluid over a non-linear stretching sheet with variable thickness, under the combined effects of thermal radiation, variable viscosity, viscous dissipation, Joule heating, and variable heat flux. This will serve as a good addition to the literature on the flow of nanofluids over non-linear stretching surfaces.

## 2. Problem Formulation

In the presence of an electric field, we investigated the 2D MHD flow of a nanofluid across a non-linear stretchable surface with changing thickness. The surface is stretched with velocity $U_{w}\left(x\right)=b{\left(x+c\right)}^{n}$, where b is a constant, c is the sheet extension parameter and n is the velocity exponential factor. For the current analysis, a rectangular coordinate system with (x,y)-axes parallel and normal to the stretching surface has been used. We suppose that the surface is not flat; therefore, its thickness varies as $y=N{\left(x+c\right)}^{\frac{1-n}{2}}$, where N is a minor factor that ensures the sheet is thin enough to avoid a noticeable pressure difference along its length. A sheet is extended from a slit by two equal and opposite forces, with the flow being regarded as going in the direction of the applied magnetic and electric field. Ohm’s law states that $J=\sigma \left(E+V\times B\right)$, where J is the current density, σ is the electrical conductivity and V is the velocity of the fluid. The magnetic field $B\left(x\right)=B_{0}{\left(x+c\right)}^{\frac{n-1}{2}}$ and electric field $E\left(x\right)=E_{0}{\left(x+c\right)}^{\frac{n-1}{2}}$ are applied normally to the flow in order to keep the Reynolds’ number as low as possible. When the magnetic Reynolds’ number is small, the generated magnetic field is less than the applied magnetic field. As a result, no induced magnetic field exists. The temperature of the surface is taken as $T_{w}=T_{\infty }+T_{0}{\left(x+c\right)}^{\frac{1-n}{2}}.$ We assume that there is a smaller temperature difference inside the flowing fluid, allowing $T^{4}$ to be stated as a linear function of temperature by using the Taylor series technique to extend $T^{4}$ at an ambient temperature $T_{\infty }$. Additionally, consideration has been given to the effects of viscous dissipation, Joule heating and non-uniform heat flux that occur in the energy equation. The geometry of the flow problem is presented in Figure 1.

Based on these assumptions, the equations for boundary-layer flow, which are made up of a continuity equation, momentum equation and energy equation, are as follows:(1)$$\frac{\partial u}{\partial x}+\frac{\partial v}{\partial y}=0,$$
(2)$$\rho_{nf}\left(u\frac{\partial u}{\partial x}+v\frac{\partial u}{\partial y}\right)=\frac{\partial }{\partial y}\left(\mu \left(T\right)\frac{\partial u}{\partial y}\right)+\sigma_{nf}\left(E\left(x\right)B\left(x\right)-B{\left(x\right)}^{2}u\right),$$
(3)$${\left(\rho c_{p}\right)}_{nf}\left(u\frac{\partial T}{\partial x}+v\frac{\partial T}{\partial y}\right)=\kappa_{nf}\frac{\partial^{2}T}{\partial y^{2}}+\frac{16\sigma^{*}T_{\infty }{}^{3}}{3k^{*}}\frac{\partial^{2}T}{\partial y^{2}}+\mu_{nf}{\left(\frac{\partial u}{\partial y}\right)}^{2}+q^{\prime \prime \prime }+\sigma_{nf}{\left(uB\left(x\right)-E\left(x\right)\right)}^{2},$$
with the following boundary conditions:(4)$$\begin{array}{l} u=U_{w}\left(x\right)=b{\left(x+c\right)}^{n},v=0,\;-\kappa \frac{\partial T}{\partial y}=h\left(T_{w}-T\right)\;\mathrm{at}\;y=N{\left(x+c\right)}^{\frac{n-1}{2}}\\ u\to 0,\;T\to T_{\infty }\;\mathrm{as}\;y\to \infty . \end{array}$$
where u and v denote the velocity components, T represents the fluid temperature, $\rho_{nf},$
$\sigma_{nf},$
$\alpha_{nf}=\frac{\kappa_{nf}}{{\left(\rho c_{{}_{p}}\right)}_{nf}}$ denotes the density, electrical conductivity and thermal diffusivity of the nanofluid. $\sigma^{*}\;\mathrm{and}\;k^{*}$ denote the Stefan–Boltzmann constant and mean absorption coefficient, respectively. The fluid’s viscosity co-efficient is defined as follows [57]:(5)$$\mu \left(T\right)=\mu^{*}\left[a_{1}+b_{1}\left(1-\theta \right)\left(T_{w}-T_{\infty }\right)\right].$$
where $a_{1}\;\mathrm{and}\;b_{1}$ are constants and $\mu^{*}$ denotes the reference viscosity. In Equation (3), the term $q^{\prime \prime \prime }$ represents the non-uniform heat flux and is expressed by the following equation [37]:(6)$$q^{\prime \prime \prime }=\frac{\kappa_{nf}U_{w}\left(x\right)}{\upsilon_{f}\left(x+c\right)}\left[A^{*}\left(T_{w}-T_{\infty }\right)+B^{*}\left(T-T_{\infty }\right)\right].$$

Here, $A^{*}\;\mathrm{and}\;B^{*}$ are space- and time-dependent heat flux parameters. Therefore, using the similarity transformations [57], the following equations can be obtained:(7)$$\begin{array}{l} u=b{\left(x+c\right)}^{n}F^{\prime }\left(\xi \right),v=-\sqrt{\frac{\left(n+1\right)\upsilon b{\left(x+c\right)}^{n-1}}{2}}\left(F\left(\xi \right)+\xi F^{\prime }\left(\xi \right)\left(\frac{n-1}{n+1}\right)\right),\\ \xi =\sqrt{\frac{\left(n+1\right)b{\left(x+c\right)}^{n-1}}{2\upsilon }}y,T=T_{\infty }+\left(T_{w}\left(x\right)-T_{\infty }\right)\Theta \left(\xi \right) \end{array}$$

Using Equations (5)–(7), the continuity equation is identically satisfied and Equations (2) and (3) take the following form:(8)$$A_{0}\left[\left(a_{1}+A\left(1-\Theta \right)\right)F^{\prime \prime \prime }-A\Theta^{\prime }F^{\prime \prime }\right]+A_{1}\left[F^{\prime \prime }F-\frac{2n}{n+1}{\left(F^{\prime }\right)}^{2}\right]+A_{2}M\left(E_{1}-F^{\prime }\right)=0,$$
(9)$$\begin{array}{l} \Theta^{\prime \prime }\left(A_{4}+\frac{4}{3}Rd\right)+\mathrm{Pr}\left[A_{0}Ec{\left(F^{\prime \prime }\right)}^{2}+A_{2}EcM{\left(F^{\prime }-E_{1}\right)}^{2}+A_{3}F\Theta^{\prime }-A_{3}F^{\prime }\Theta \left(\frac{1-n}{1+n}\right)\right]+\\ A_{4}\frac{2}{n+1}\left(A^{*}F^{\prime }+B^{*}\Theta \right)=0 \end{array}$$

The boundary conditions in Equation (4) are given as
(10)$$\begin{array}{l} F\left(\xi \right)=\left(\frac{1-n}{1+n}\right)\alpha ,F^{\prime }\left(\xi \right)=1,\Theta^{\prime }\left(\xi \right)=-\delta \left(1-\Theta (0)\right)\;\;\mathrm{at}\;\xi =0\\ \;F^{\prime }\left(\xi \right)=0,\Theta \left(\xi \right)=0\;\mathrm{as}\;\xi \to \infty \end{array}$$

According to Ref. [57],
(11)$$\begin{array}{l} F\left(\xi \right)=f\left(\xi -\alpha \right)=f\left(\eta \right),\\ \Theta \left(\xi \right)=\theta \left(\xi -\alpha \right)=\theta \left(\eta \right) \end{array}$$

Equations (8)–(10) become
(12)$$A_{0}\left[\left(a_{1}+A\left(1-\theta \right)\right)f^{\prime \prime \prime }-A\theta^{\prime }f^{\prime \prime }\right]+A_{1}\left[f^{\prime \prime }f-\frac{2n}{n+1}{\left(f^{\prime }\right)}^{2}\right]+A_{2}M\left(E_{1}-f^{\prime }\right)=0,$$
(13)$$\begin{array}{l} \theta^{\prime \prime }\left(A_{4}+\frac{4}{3}Rd\right)+\mathrm{Pr}\left[A_{0}Ec{\left(f^{\prime \prime }\right)}^{2}+A_{2}EcM{\left(f^{\prime }-E_{1}\right)}^{2}+A_{3}f\theta^{\prime }-A_{3}f^{\prime }\theta \left(\frac{1-n}{1+n}\right)\right]+\\ A_{4}\frac{2}{n+1}\left(A^{*}f^{\prime }+B^{*}\theta \right)=0 \end{array}$$
(14)$$\begin{array}{l} f\left(\eta \right)=\left(\frac{1-n}{1+n}\right)\alpha ,f^{\prime }\left(\eta \right)=1,\theta^{\prime }\left(\eta \right)=-\delta \left(1-\theta \left(0\right)\right)\;\;\mathrm{at}\;\eta =0\\ \;f^{\prime }\left(\eta \right)=0,\theta \left(\eta \right)=0\;\mathrm{as}\;\eta \to \infty \end{array}$$

The non-dimensional parameters that appear in Equations (7)–(9) are the magnetic field parameter M, electric field parameter $E_{1},$ radiation parameter Rd, Eckert number Ec, Prandtl number Pr, wall thickness parameter α, heat transfer Biot number δ, and dimensionless variable viscosity parameter A, which are defined as follows:(15)$$\begin{array}{l} M=\frac{2\sigma_{f}B_{0}{}^{2}}{\rho_{f}b\left(n+1\right)},E_{1}=\frac{E_{0}}{B_{0}b{\left(x+c\right)}^{n}},Rd=\frac{4\sigma^{*}T_{\infty }^{3}}{k^{*}\kappa_{f}},\;Ec=\frac{b^{2}{\left(x+c\right)}^{2n}}{{\left(c_{p}\right)}_{f}\left(T_{w}-T_{\infty }\right)},\\ \mathrm{Pr}=\frac{\upsilon_{f}{\left(\rho c_{p}\right)}_{f}}{\kappa_{f}},\alpha =N\sqrt{\frac{b\left(n+1\right)}{2\upsilon_{f}}},\delta =\frac{h}{\kappa_{f}}\sqrt{\frac{2\upsilon_{f}}{\left(n+1\right)b}{\left(x+c\right)}^{1-n}},A=b_{1}\left(T_{w}-T_{\infty }\right). \end{array}$$

## 3. Thermo-Physical Properties

Table 1 represents the experimental relationship between the nanofluid’s thermo-physical properties, where $\rho_{nf},$$\mu_{nf}$,${\left(\rho c_{p}\right)}_{nf}$, $\kappa_{nf}$ are the density, dynamic viscosity, specific heat capacity, and thermal conductivity of the nanofluid, respectively, as given by Hamilton [61] and Raza et al. [62]. ϕ is the volume fraction of nanoparticles, $\rho_{p}$ and $\rho_{f}$ are the densities of the nanoparticles and the base fluid, ${\left(\rho c_{p}\right)}_{p}$ and ${\left(\rho c_{p}\right)}_{f}$ are the heat capacities of nanoparticles and the base fluid; $\kappa_{p}$ and $\kappa_{f}$ are the thermal conductivities of the nanoparticles and base fluid. Table 2 shows the numerical values of the thermophysical quantities of the nanomaterials, which are used to calculate the values of the constant.

## 4. Physical Quantities

The following definitions provide the mathematical formulations for the wall shear stress and the heat transfer rate in terms of thermal flux at the wall:(16)$$\begin{array}{l} C_{f}=\frac{\tau_{w}}{\rho_{nf}U_{w}^{2}},\;\tau_{w}={\left.\mu_{nf}\left(\frac{\partial u}{\partial y}\right)\right|}_{y=N{\left(x+c\right)}^{\frac{n-1}{2}}}\\ \;Nu_{x}=\frac{\left(x+c\right)q_{w}}{\kappa_{f}\left(T_{w}-T_{\infty }\right)},\;q_{w}=-\kappa_{nf}{\left.\frac{\partial T}{\partial y}\right|}_{y=N{\left(x+c\right)}^{\frac{n-1}{2}}}, \end{array}$$

We obtain the following dimensionless form of the skin friction coefficient and Nusselt number by incorporating the similarity variables from Equations (7) and (11) into Equation (16) as follows:(17)Rex1/2Cf=A0A1n+12f″0,  Rex−1/2Nux=−A4n+12θ′0,
where the local Reynolds number Rex, a dimensionless quantity, is defined as Rex=Uwx+c/νf.

## 5. Numerical Method

In this part, we explored the numerical approach for solving our non-dimensionalized equations under the provided boundary conditions. The Adam–Bashforth predictor corrector method, being a linear multistep technique, was employed for this purpose. Multistep techniques try to enhance effectiveness by retaining and using previous phase information instead of deleting it; hence, they make use of such a large number of earlier points and derivative values. It works in two phases, which are as follows: firstly, we used the Adam–Bashforth technique to generate an approximation of the desired values as a predictor. Then, we used the Adam–Moulton method as a corrector step that rectifies the initial estimate. The first order system for $f\left(\eta \right)$ and $\theta \left(\eta \right)$ is as follows:$$f_{1}=f^{\prime },f_{2}={f_{1}}^{\prime },f_{3}={f_{2}}^{\prime }$$
$$f_{3}={f_{2}}^{\prime }=\frac{1}{\left[a_{1}+A\left(1-\theta \right)\right]}\left[A\theta_{1}f_{2}-\frac{A_{1}}{A_{0}}\left(f_{2}f-\frac{2n}{n+1}{\left(f_{1}\right)}^{2}\right)-\frac{A_{2}}{A_{0}}M\left(E_{1}-f_{1}\right)\right]$$
and the temperature equation $\theta \left(\eta \right)$ can be expressed as
$$\theta_{1}=\theta^{\prime },\theta_{2}={\theta_{1}}^{\prime }$$
$$\theta_{2}={\theta_{1}}^{\prime }=\frac{1}{\left(1+\frac{4}{3}Rd\right)}\left[-\mathrm{Pr}\left(\frac{A_{0}}{A_{4}}Ec{\left(f_{2}\right)}^{2}+\frac{A_{2}}{A_{4}}EcM{\left(f_{1}-E_{1}\right)}^{2}+\frac{A_{3}}{A_{4}}f\theta_{1}-\frac{A_{3}}{A_{4}}f_{1}\theta \left(\frac{1-n}{1+n}\right)\right)\right]$$

The suitable boundary conditions for $f\left(\eta \right)$ and $\theta \left(\eta \right)$ are given as follows:$$\begin{array}{l} f\left(0\right)=\left(\frac{1-n}{1+n}\right)\alpha ,f^{\prime }\left(0\right)=1,\\ \theta^{\prime }\left(0\right)=-\delta \left(1-\theta \left(0\right)\right) \end{array}$$

In the above equations,
$$\begin{array}{l} A_{0}=\frac{\mu_{nf}}{\mu_{f}}=\frac{1}{{\left(1-\phi \right)}^{2.5}},\\ A_{1}=\frac{\rho_{nf}}{\rho_{f}}=\left(1-\phi \right)+\phi \frac{\rho_{p}}{\rho_{f}},\\ A_{2}=\frac{\sigma_{nf}}{\sigma_{f}}=1+\frac{3\phi \left(\frac{\sigma_{p}}{\sigma_{f}}-1\right)}{\left(\frac{\sigma_{p}}{\sigma_{f}}+2\right)-\left(\frac{\sigma_{p}}{\sigma_{f}}-1\right)\phi },\\ A_{3}=\frac{{\left(\rho c_{p}\right)}_{nf}}{{\left(\rho c_{p}\right)}_{f}}=\left(1-\phi \right)+\phi \frac{{\left(\rho c_{p}\right)}_{p}}{{\left(\rho c_{p}\right)}_{f}},\\ A_{4}=\frac{\kappa_{nf}}{\kappa_{f}}=\left(\frac{2\kappa_{f}+\kappa_{p}-2\phi \left(\kappa_{f}-\kappa_{p}\right)}{2\kappa_{f}+\kappa_{p}+\phi \left(\kappa_{f}-\kappa_{p}\right)}\right). \end{array}$$

The acquired differential system for $f\left(\eta \right)$ and $\theta \left(\eta \right)$ are commonly expressed as follows:$$\begin{array}{l} \frac{df}{d\eta }=q\left(\eta ,f\right),\;f\left(\eta_{0}\right)=f_{0},\\ \frac{d\theta }{d\eta }=q\left(\eta ,\theta \right),\;\theta \left(\eta_{0}\right)=\theta_{0}. \end{array}$$

The Adam–Bashforth predictor approach can be expressed in general as
$$\begin{array}{l} f_{k+1}=f_{k}+\frac{3h}{2}q\left(\eta_{k},f_{k}\right)-\frac{h}{2}q\left(\eta_{k-1},f_{k-1}\right)\\ \theta_{k+1}=\theta_{k}+\frac{3h}{2}q\left(\eta_{k},\theta_{k}\right)-\frac{h}{2}q\left(\eta_{k-1},\theta_{k-1}\right), \end{array}$$

The general expression for the Adam–Moulton corrector approach is as follows:$$\begin{array}{l} f_{k+1}=f_{k}+\frac{h}{2}q\left(\eta_{k+1},f_{k+1}\right)-\frac{h}{2}q\left(\eta_{k},f_{k}\right),\\ \theta_{k+1}=\theta_{k}+\frac{h}{2}q\left(\eta_{k+1},\theta_{k+1}\right)-\frac{h}{2}q\left(\eta_{k},\theta_{k}\right) \end{array}$$
where *h* is a parameter that defines the step size.

For the validation of the results obtained from the Adam–Bashforth predictor-corrector technique, we have compared our results with those obtained from the explicit Runge–Kutta method. In Figure 2a, we plot the velocity profile obtained from both the Adam–Bashforth method and the explicit Runge–Kutta method for varying values of the wall thickness parameter α. In Figure 2b, we present the absolute error for different values of α. It is observed that the absolute error between the results obtained from the Adam–Bashforth method and the explicit Runge–Kutta method is around $10^{-6}-10^{-10}$, which is considered to be negligible, and hence validates our results. Similarly, Figure 3a,b show the temperature profile and absolute error for varying values of the Hartmann number M.

## 6. Graphical Analysis

In this portion, we have discussed the effects and behavior of physical characteristics on all the profiles. The velocity and temperature curves for numerous physical parameter values are shown in Figure 4, Figure 5, Figure 6, Figure 7, Figure 8, Figure 9, Figure 10 and Figure 11. These different parameters include magnetic and electric field parameters; the wall thickness parameter; radiation parameter; viscous dissipation parameter; velocity power index; heat transfer Biot number; and heat generation/absorption parameter. The alteration of the velocity power index n determines the behavior of the slandering surface. The impact of n on the velocity and temperature profiles is depicted in Figure 4. An increase in n values enhances the stretching velocity, causing more fluid deformation, which, as a result, causes the elevation of the velocity boundary layer. In addition, the temperature of the fluid raises with n, causing the thickness of the thermal layer to increase.

The impact of the wall thickness parameter α on velocity and temperature is presented in Figure 5. The flow characteristics are considerably reduced when they approach the surface, resulting in a decrease in wall thickness in accordance with the velocity for each position inside this zone. The temperature gradient is also depreciated by the wall thickness parameter. This is due to the lower thermal energy transfer through the thicker parts towards the fluid flow, as compared to the more narrow parts.

Figure 6 shows the relation between the magnetic field parameter M with fluid velocity and temperature. It was found that boosting the magnetic field lessens the movement of nanoparticles. The impact of a vertical magnetic field on an electrically conductive nanoparticle induces an escalation in the Lorentz force, which is a type of resistive force. The magnetism property can be used to manipulate fluid velocity. As a result, the increment of M enables the fluid velocity to descend. Similarly, due to an increment in the values of M, the velocity gradient drops, which results in the formation of Lorentz force, thus amplifying the boundary-layer thickness and process of heat conduction, leading to a rise in thermal distribution.

Figure 7 shows the effect of temperature distribution on the viscous dissipation parameter. The Eckert number illustrates heat dissipation by showing the relationship between the fluid flow’s kinetic energy and the difference in enthalpy between the boundary layer and the surrounding medium. As the Eckert number rises, the temperature difference between the fluid’s surface and the surrounding air reduces, allowing the fluid’s temperature to rise across the domain. As a result, the thickness of the thermal boundary layer is typically enhanced.

The temperature profile for various Rd values can be observed in Figure 8. The rate of conduction heat transfer to thermal radiation transfer is described by the radiation parameter. Thermal radiation accelerates heat transfer because the enhancement in thermal radiation will increase the thickness of the thermal boundary layer. The temperature and the thickness of the thermal boundary layer rise as a result of the higher values of Rd supplying more heat to the working fluid. The heat transfer Biot number δ generates heat, which is displayed in Figure 9. The Biot number is expressed as the ratio of the convective heat transfer coefficient to heat conduction and is associated with the convective boundary conditions somewhere at the surface. An increment in its value represents a high temperature at the surface, decreasing fluid conductance and raising the thickness of the temperature profile.

The consequence of the space-dependent heat generation/absorption factor $A^{*}$ can be observed in Figure 10a,b. It was revealed that the temperature distribution engenders energy, enabling the temperature gradient to expand, due to the escalation in the values of $A^{*}$, whereas the boundary layer absorbs heat, which leads to a substantial drop in temperature for the negative values of $A^{*}.$ The effects of a time-dependent heat generation/absorption factor on temperature are shown in Figure 11a,b. These graphs reflect that as the values of $B^{*}$ increase, i.e., $B^{*}>0$, more heat energy is produced. As a result of this, the temperature rises. When the value of $B^{*}$ decreases, however, more heat energy is absorbed, resulting in a significant decrease in temperature.

Table 3 compares the results for the limiting case to those found in the literature by Daniel et al. [33]. We can observe that the values of $-f^{\prime \prime }\left(0\right)$ for the changing values of n show good agreement.

## 7. Conclusions

In this work, we have investigated a two-dimensional EMHD nanofluid flow on a stretchable surface with varying thickness. The effects of magnetic fields, electric fields, variable heat flux, variable fluid viscosity, thermal radiation, viscous dissipation, and Joule heating on a stretching surface of variable thickness have all been explored. The leading equations after using similarity transformations and non-dimensional variables were computed numerically. By visualizing velocity and temperature distributions, the impacts of various physical parameters were investigated. The main observations drawn from this study are as follows:The fluid’s velocity, as well as the temperature of the nanofluid, increase when the velocity power index increases.By increasing the wall thickness parameter, the fluid’s velocity and temperature are reduced.As the temperature rises, both the fluid velocity decreases and the Hartmann number decrease.By raising the heat transfer Biot number, thermal radiation parameter, and viscous dissipation parameter, the fluid’s temperature rises.When the positive values increase, the impact of the space-dependent and time-dependent heat generation and absorption parameters generates heat and absorbs temperature when the negative values decrease.

## Figures and Tables

**Figure 1 nanomaterials-12-03872-f001:**
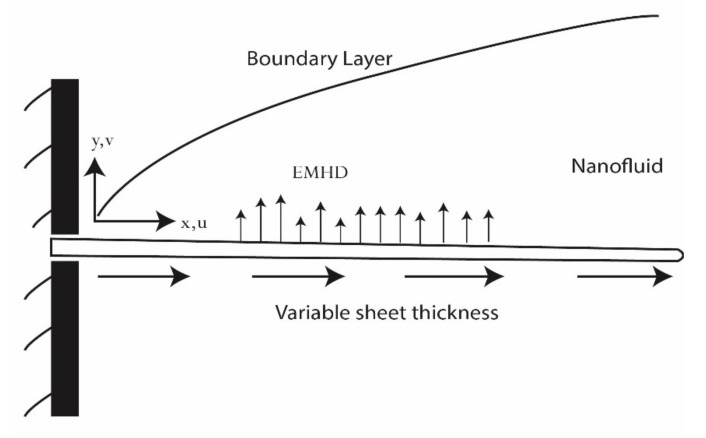
Schematic diagram of the model.

**Figure 2 nanomaterials-12-03872-f002:**
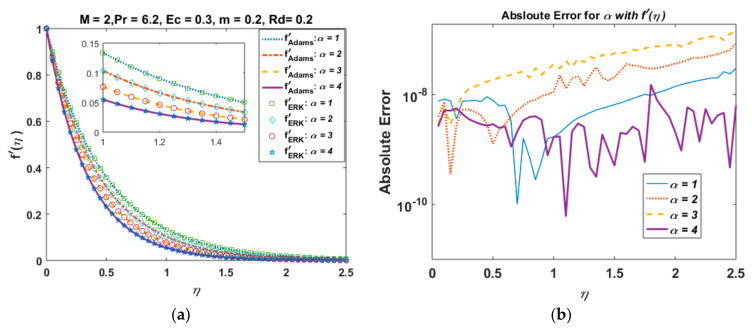
Velocity profile and absolute error for varying values of α. (**a**) Velocity profile with Adam–Bashforth and explicit Runge–Kutta method. (**b**) Absolute error.

**Figure 3 nanomaterials-12-03872-f003:**
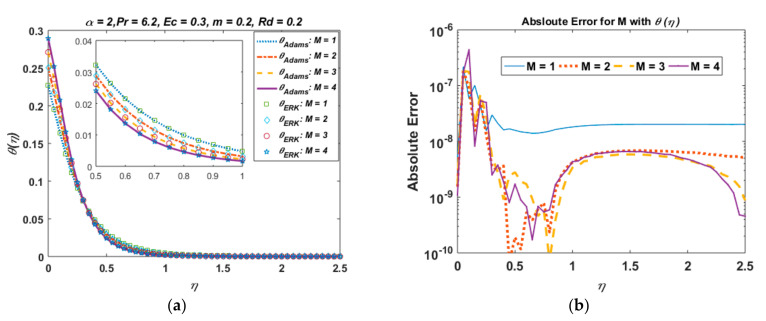
Temperature profile and absolute error for varying values of M. (**a**) Temperature profile with Adam–Bashforth and explicit Runge–Kutta method. (**b**) Absolute error.

**Figure 4 nanomaterials-12-03872-f004:**
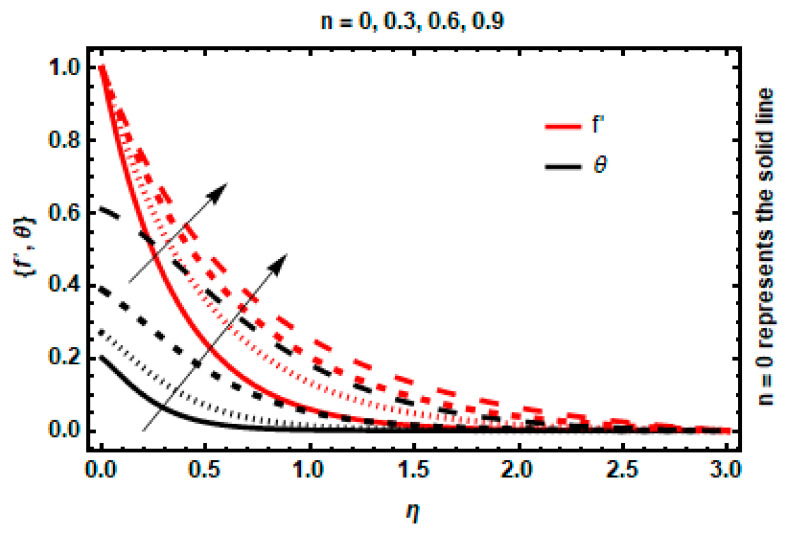
Impact of *n* on velocity and temperature.

**Figure 5 nanomaterials-12-03872-f005:**
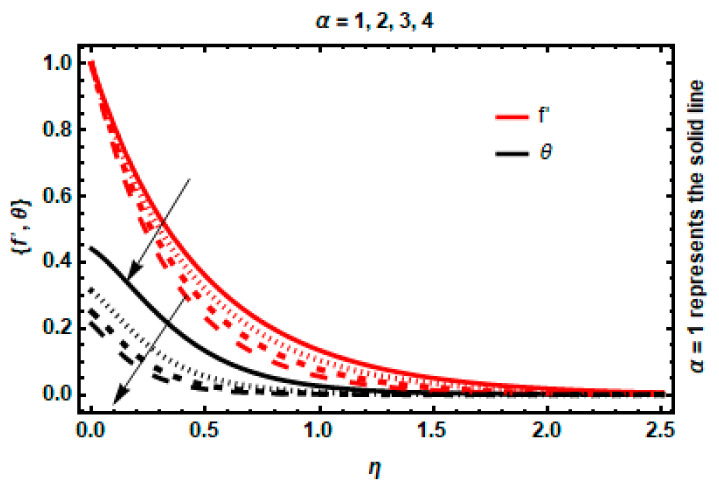
Impact of α on velocity and temperature.

**Figure 6 nanomaterials-12-03872-f006:**
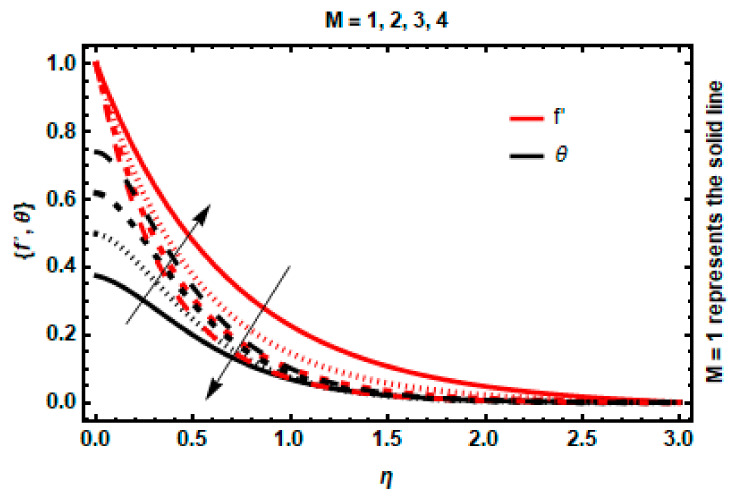
Impact of *M* on velocity and temperature.

**Figure 7 nanomaterials-12-03872-f007:**
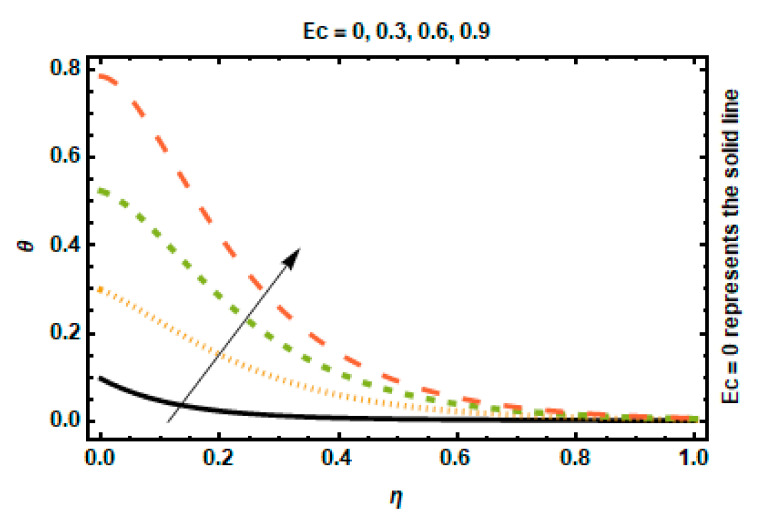
Effects of *Ec* on temperature.

**Figure 8 nanomaterials-12-03872-f008:**
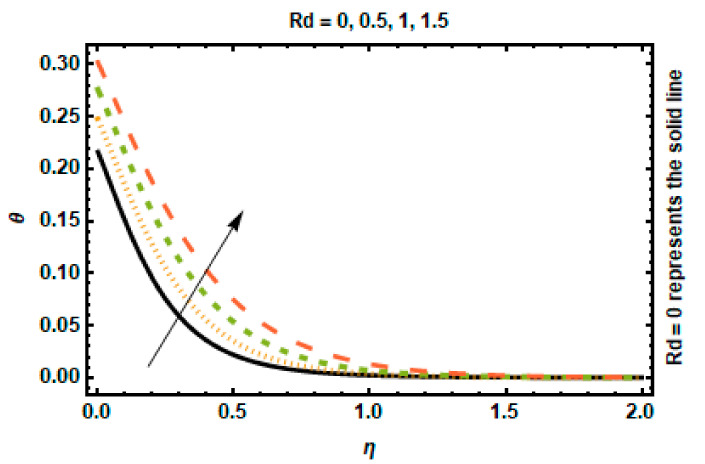
Effects of *Rd* on temperature.

**Figure 9 nanomaterials-12-03872-f009:**
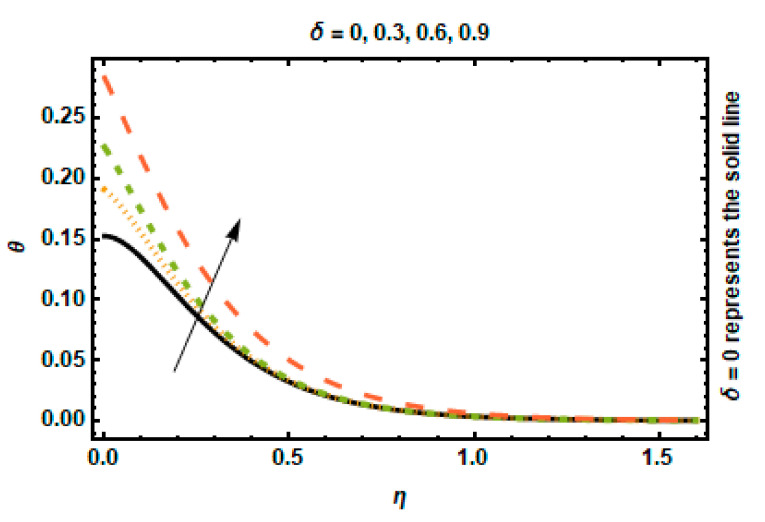
Effects of *δ* on temperature.

**Figure 10 nanomaterials-12-03872-f010:**
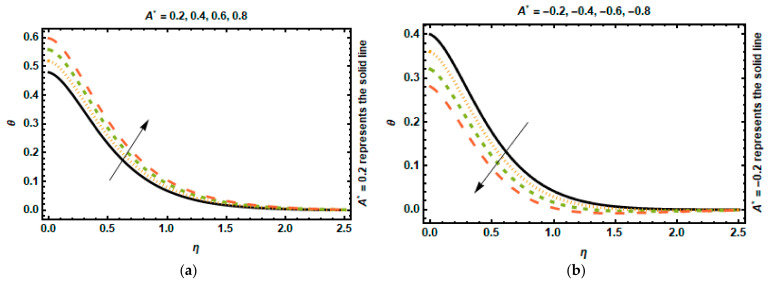
Effects of $A^{*}$ on temperature. (**a**) $A^{*}>0$. (**b**) $A^{*}<0$.

**Figure 11 nanomaterials-12-03872-f011:**
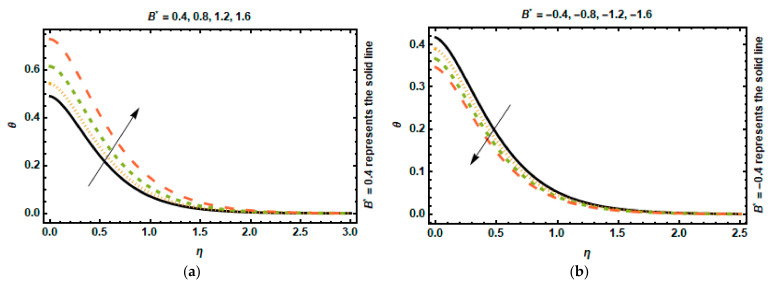
Effects of $B^{*}$ on temperature. (**a**) $B^{*}>0$. (**b**) $B^{*}<0$.

**Table 1 nanomaterials-12-03872-t001:** Thermal properties of nanofluids.

Properties	Nanofluid
Density	$\rho_{nf}=(1-\phi )\rho_{f}+\phi \rho_{p}$
Viscosity	$\mu_{nf}=\frac{\mu_{f}}{{\left(1-\phi \right)}^{2.5}}$
Heat capacity	${\left(\rho c_{p}\right)}_{nf}=\left(1-\phi \right){\left(\rho c_{p}\right)}_{f}+\phi {\left(\rho c_{p}\right)}_{p}$
Thermal conductivity	$\kappa_{nf}=\left(\frac{\kappa_{p}+2\kappa_{f}-2\phi \left(\kappa_{f}-\kappa_{p}\right)}{\kappa_{p}+2\kappa_{f}+\phi \left(\kappa_{f}-\kappa_{p}\right)}\right)\kappa_{f}$
Electric conductivity	$\sigma_{nf}=1+\frac{3\phi \left(\frac{\sigma_{p}}{\sigma_{f}}-1\right)}{\left(\frac{\sigma_{p}}{\sigma_{f}}+2\right)-\left(\frac{\sigma_{p}}{\sigma_{f}}-1\right)\phi }\sigma_{f}$

**Table 2 nanomaterials-12-03872-t002:** Numerical values of thermo-physical properties of nanoparticles and base fluid.

Physical Properties	Base Fluid (H_2_O)	Copper (Cu)
$\rho \left({\mathrm{kg}/m}^{3}\right)$	997.1	8933
$c_{p}\left(J/\mathrm{kg}\cdot K\right)$	4179	385
κ(W/mk)	0.613	401
$\sigma \left(/\Omega m\right)$	0.05	$5.96\times 10^{7}$

**Table 3 nanomaterials-12-03872-t003:** Comparison of $-f^{\prime \prime }\left(0\right)$ when $E_{1}=0$,M=0, A=0, $a_{1}=1$, ϕ=0, and α=0.25 for changing values of n.

n	Daniel et al. [33]	Present Result
10	1.143316	1.143220
9	1.140388	1.140575
7	1.132281	1.132588
5	1.118587	1.118767
3	1.090490	1.090916
1	1.000001	1.000019
0.5	0.933828	0.933556
0	0.784284	0.784312
−1/3	0.500000	0.500003
−0.5	0.083289	0.083242
−0.51	0.038484	0.038349
−0.55	−0.197647	−0.197772
−0.60	−0.850207	−0.850697
−0.61	−1.224426	−1.22459

## Data Availability

Not applicable.

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
