# Peer review of "EMHD Nanofluid Flow with Radiation and Variable Heat Flux Effects along a Slandering Stretching Sheet"

_nanomaterials, 2022, doi:10.3390/nano12213872_

Round 1
Reviewer 1 Report
EMHD nanofluid flow with radiation and variable heat flux effects along a Slandering stretching sheet In this article the authors studied the heat transfer analysis of EMHD nanofluid flow over a non-linear stretching surface with variable thickness. They consider the nanometer-sized Copper particles in the base fluid which is taken water. The governing equations has been simplified by using similarity parameters and non-dimensional variables and then solved numerically. The results are explained graphically with detail physical reasons. The effects of nanofluid are analyzed in detail and found very interesting results. In my opinion the article is suitable for publication in this Journal. My overall recommendation is in favor to publish the paper in “Nanomaterials”. However, there are some points which need to be addressed: 1. The introduction section should be expanded with the help of more recent papers on the topics. 2. All grammatical and typo mistakes should be removed. 3. Expressions for Skin friction and rate of heat transfer should be computed. 4. Adding SI unites to nomenclature is recommended. 5. Figures are of low quality which cause difficulty in reading. Improve the quality of figures.
Author Response
Comments and Suggestions for Authors:
EMHD nanofluid flow with radiation and variable heat flux effects along a Slandering stretching sheet. In this article the authors studied the heat transfer analysis of EMHD nanofluid flow over a non-linear stretching surface with variable thickness. They consider the nanometer-sized Copper particles in the base fluid which is taken water. The governing equations has been simplified by using similarity parameters and non-dimensional variables and then solved numerically. The results are explained graphically with detail physical reasons. The effects of nanofluid are analyzed in detail and found very interesting results. In my opinion the article is suitable for publication in this Journal. My overall recommendation is in favor to publish the paper in “Nanomaterials”. However, there are some points which need to be addressed:
- The introduction section should be expanded with the help of more recent papers on the topics.
Response: As suggested the introduction section has been expanded by adding recent papers in the revised manuscript.
- All grammatical and typo mistakes should be removed.
Response: As suggested all the typo and grammatical errors have been removed in the revised manuscript.
- Expressions for Skin friction and rate of heat transfer should be computed.
Response: As suggested the expressions for skin friction and Nusselt number have been computed in the revised manuscript.
- Adding SI unites to nomenclature is recommended.
Response: As suggested the nomenclature has been updated by adding SI units in the revised manuscript.
- Figures are of low quality which cause difficulty in reading. Improve the quality of figures.
Response: As suggested the quality of the figures has been improved in the revised manuscript.
Reviewer 2 Report
Title:
“EMHD nanofluid flow with radiation and variable heat flux effects along a Slandering stretching sheet"
In general, the subject is interesting and could be of practical application. The quality of the writing is needed to improve. The wrong structure, as well as lousy punctuation in some sentences, prevents proper understanding. Revision must be checked before the decision. Here are some suggestions that must be considered to improve the quality of work.
- The abstract should contain answers to the following questions: What problem was studied and why is it important? What methods were used? What are the important results? What conclusions can be drawn from the results? What is the novelty of the work and where does it go beyond previous efforts in the literature? Please include specific and quantitative results in your abstract, while ensuring that it is suitable for a broad audience.
- I suggest the novelty should be emphasized in a more clear way including the comparison of the literature review and also mentioning present work motivation.
- The last paragraph of the introduction should be clearer, what you have done, where was the research gap? How you have fulfilled it? How do you imagine that your work is applicable in which field, mentioned its application?
- Grammatical language is very poor, authors must improve it. The authors should revise all the manuscripts.
- All acronyms must be defined at first appearance in the abstract and again in the manuscript, it should be clear to readers.
- The introduction section should be improve and should be according to keywords, be in a sequence and correlated and decorate with the latest studies. doi.org/10.1016/j.csite.2022.102247; https://doi.org/10.1016/j.camwa.2022.01.009.
- There is a mistake in equ.9, make corrections, also in the next equations and graphs. Why the authors have used similarity transformations?? Is it necessary to use? The problem can be solved in the PDEs system. Could I ask not to use similarity transformations?
- The result and discussion part needs to be improve and more physical reasons must be included.
- What is the difference between EMHD and MHD? How you have differentiated flow figures and problem descriptions?
- Why you did not consider the continuity equation? How you can prove that your model is valid?
- The authors should provide the results comparison table. Also, the graphs should zoom out because the results are not clear.
- Improve the conclusion part and describe the main findings and its applications.
Author Response
Comments and Suggestions for Authors:
Title: “EMHD nanofluid flow with radiation and variable heat flux effects along a Slandering stretching sheet"
In general, the subject is interesting and could be of practical application. The quality of the writing is needed to improve. The wrong structure, as well as lousy punctuation in some sentences, prevents proper understanding. Revision must be checked before the decision. Here are some suggestions that must be considered to improve the quality of work.
- The abstract should contain answers to the following questions: What problem was studied and why is it important? What methods were used? What are the important results? What conclusions can be drawn from the results? What is the novelty of the work and where does it go beyond previous efforts in the literature? Please include specific and quantitative results in your abstract, while ensuring that it is suitable for a broad audience.
Response: As suggested we have update the abstract by including all the suggestions in the revised manuscript.
- I suggest the novelty should be emphasized in a more clear way including the comparison of the literature review and also mentioning present work motivation.
Response: As suggested we have improve the literature review of the revised manuscript by adding more recent papers which clearly shows the gap between the literature work and the current study. By comparing the literature work with the current work we have clearly mentioned the novelty of this work in the introduction section.
- The last paragraph of the introduction should be clearer, what you have done, where was the research gap? How you have fulfilled it? How do you imagine that your work is applicable in which field, mentioned its application?
Response: As suggested we have clearly mentioned the aim of this work in the last paragraph of the introduction section of the revised manuscript.
- Grammatical language is very poor, authors must improve it. The authors should revise all the manuscripts.
Response: As suggested we have improve the language of the revised manuscript by removing all the grammatical and typo errors.
- All acronyms must be defined at first appearance in the abstract and again in the manuscript,it should be clear to readers.
Response: As suggested we have clearly define all the acronyms appeared in the manuscript in nomenclature section of the revised manuscript.
- The introduction section should be improveand should be according to keywords, be in a sequence and correlated and decorate with the latest studies.
https://doi.org/10.1016/j.csite.2022.102247;
https://doi.org/10.1016/j.camwa.2022.01.009.
Response: As suggested the introduction section of the revised manuscript has been updated by adding the updated references (See [35], [45]).
- There is a mistake in equ.9, make corrections, also in the next equations and graphs. Why the authors have used similarity transformations?? Is it necessary to use? The problem can be solved in the PDEs system. Could I ask not to use similarity transformations?
Response: We have corrected Eq. (9) and its subsequent equation, it is a typo mistake. We have use similarity transformations which reduces our system of nonlinear partial differential equations to a system of nonlinear ordinary differential equations which are relatively easy to solve. Further, it reduces the number of variables as well as the number of equations since the continuity equation is identically satisfied by using the similarity transformation. We have adopted this procedure but yes it is not necessary to use these similarity transformations and partial differential equations can be solved.
- The result and discussion part needs to be improve and more physical reasons must be included.
Response: As suggested we have improve the result and discussion part by adding more physical description of the parameters in the revised manuscript.
- What is the difference between EMHD and MHD? How you have differentiated flow figures and problem descriptions?
Response: The difference between EMHD and MHD has been defined in the introduction section of the revised manuscript.
- Why you did not consider the continuity equation? How you can prove that your model is valid?
Response: We have consider the continuity equation (1). When we use the similarity transformations (7) into continuity equation (1), then it is identically satisfied. This is one of the advantages of using the similarity transformations that it reduces the number of variables as well as the number of equations.
- The authors should provide the results comparison table. Also, the graphs should zoom out because the results are not clear.
Response: As suggested we have present a comparison table in which numerical values are compared as a special case of the previous study available in the literature.
- Improve the conclusion part and describe the main findings and its applications.
Response: As suggested we have improve the conclusion section in the revised manuscript.
Reviewer 3 Report
The paper "EMHD nanofluid flow with radiation and variable heat flux effects along a Slandering stretching sheet" deals with an interesting subject, but it needs some revisions:
1) A nomenclature should be added in order to list all the symbols and the corresponding units of measure.
2) A deeper bibliographic analysis is required in order to give a complete understanding of the state of the art. It is suggested to add these papers (and more):
Stagnation electrical MHD nanofluid mixed convection with slip boundary on a stretching sheet, Applied Thermal Engineering, Volume 98, Pages 850 - 861
Effects of radiative electro-magnetohydrodynamics diminishing internal energy of pressure-driven flow of titanium dioxide-water nanofluid due to entropy generation, Entropy, Volume 21, Issue 3
Experimental Measurements of Al2O3 and CuO Nanofluids Interaction with Microwaves, 2017, Journal of Energy Engineering, 143(2),04016045
Numerical spectral examination of EMHD mixed convective flow of second-grade nanofluid towards a vertical Riga plate using an advanced version of the revised Buongiorno’s nanofluid model, Journal of Thermal Analysis and CalorimetryVolume 143, Issue 3, Pages 2379 – 2393
Effects of thermal radiation and electromagnetohydrodynamics on viscous nanofluid through a Riga plate, Multidiscipline Modeling in Materials and StructuresVolume 12, Issue 4, Pages 605 - 618
A critical analysis of clustering phenomenon in Al2O3 nanofluids, 2019, Journal of Thermal Analysis and Calorimetry 135(1), pp. 371-377, DOI: 10.1007/s10973-018-7099-9
Critical Review of Experimental Investigations about Convective Heat Transfer Characteristics of Nanofluids under Turbulent and Laminar Regimes with a Focus on the Experimental Setup. Energies 2021, 14, 6004. https://doi.org/10.3390/en14186004;
Heat Transfer Attributes of Gold–Silver–Blood Hybrid Nanomaterial Flow in an EMHD Peristaltic Channel with Activation Energy, NanomaterialsOpen AccessVolume 12, Issue 10May-2 2022 Article number 1615
3) The experimental validation is very limited and it is not general. There is not an error analysis of the results.
4) More details should be given on how the thermophysical properties of the nanofluids are accounted for.
5) More comments explaining the physical effects which determined the obtained results are required.
6) The authors could group the plots in less images, in order to reduce the number of pictures
7) It is not clear the advantage of the mathematical model and its scientific relevance due to its intrinsic limitations.
8) Hence, I believe that the authors have to redraft the manuscript with added discussions providing significant physical insights.
Author Response
Comments and Suggestions for Authors:
The paper "EMHD nanofluid flow with radiation and variable heat flux effects along a Slandering stretching sheet" deals with an interesting subject, but it needs some revisions:
1) A nomenclature should be added in order to list all the symbols and the corresponding units of measure.
Response: As suggested we have update the nomenclature by adding the complete list of symbols and their corresponding units in the revised manuscript.
2) A deeper bibliographic analysis is required in order to give a complete understanding of the state of the art. It is suggested to add these papers (and more):
Stagnation electrical MHD nanofluid mixed convection with slip boundary on a stretching sheet, Applied Thermal Engineering, Volume 98, Pages 850 – 861.
Effects of radiative electro-magnetohydrodynamics diminishing internal energy of pressure-driven flow of titanium dioxide-water nanofluid due to entropy generation, Entropy, Volume 21, Issue 3.
Experimental Measurements of Al2O3 and CuO Nanofluids Interaction with Microwaves, 2017, Journal of Energy Engineering, 143(2), 04016045.
Numerical spectral examination of EMHD mixed convective flow of second-grade nanofluid towards a vertical Riga plate using an advanced version of the revised Buongiorno’s nanofluid model, Journal of Thermal Analysis and Calorimetry Volume 143, Issue 3, Pages 2379 – 2393.
Effects of thermal radiation and electromagnetohydrodynamics on viscous nanofluid through a Riga plate, Multidiscipline Modeling in Materials and Structures Volume 12, Issue 4, Pages 605 – 618.
A critical analysis of clustering phenomenon in Al2O3 nanofluids, 2019, Journal of Thermal Analysis and Calorimetry 135(1), pp. 371-377, DOI: 10.1007/s10973-018-7099-9.
Critical Review of Experimental Investigations about Convective Heat Transfer Characteristics of Nanofluids under Turbulent and Laminar Regimes with a Focus on the Experimental Setup. Energies 2021, 14, 6004. https://doi.org/10.3390/en14186004.
Heat Transfer Attributes of Gold–Silver–Blood Hybrid Nanomaterial Flow in an EMHD Peristaltic Channel with Activation Energy, Nanomaterials Open Access Volume 12, Issue 10May-2 2022 Article number 1615.
Response: As suggested the introduction section of the revised manuscript has been updated by adding the updated references (See [17], [19], [27], [28], [36], [46], [53]).
3) The experimental validation is very limited and it is not general. There is not an error analysis of the results.
Response: As suggested we have include an error analysis to validate our obtained results in the revised manuscript.
4) More details should be given on how the thermophysical properties of the nanofluids are accounted for.
Response: As suggested we have given more about the thermophysical properties of the nanofluid by defining all the properties and adding the relevant reference in the revised manuscript.
5) More comments explaining the physical effects which determined the obtained results are required.
Response: As suggested more physical description has been added in the revised manuscript.
6) The authors could group the plots in less images, in order to reduce the number of pictures.
Response: As suggested we have group the plots and the number of figures has been reduced in the revised manuscript.
7) It is not clear the advantage of the mathematical model and its scientific relevance due to its intrinsic limitations.
Response: As suggested we have clearly mentioned the advantages of the mathematical model in the introduction section of the revised manuscript.
8) Hence, I believe that the authors have to redraft the manuscript with added discussions providing significant physical insights.
Response: As suggested we have improve the manuscript draft by removing all the grammatical and typo errors and add more physical insights.
Round 2
Reviewer 2 Report
Agreed
Reviewer 3 Report
The Authors revise the paper properly.